# Sampled-Data Exponential Synchronization of Complex Dynamical Networks with Saturating Actuators

**DOI:** 10.3390/e26090785

**Published:** 2024-09-14

**Authors:** Runan Guo, Wenshun Lv

**Affiliations:** 1College of Electrical Engineering and Automation, Shandong University of Science and Technology, Qingdao 266590, China; flyguorn@163.com; 2School of Science, Qingdao University of Technology, Qingdao 266520, China

**Keywords:** complex dynamical networks, synchronization, sampled-data control, actuator saturation

## Abstract

This paper investigates the problem of exponential synchronization control for complex dynamical systems (CDNs) with input saturation. Considering the effects of transmission delay, a memory sampled-data controller is designed. A modified two-sided looped functional is constructed that takes into account the entire sampling period, which includes both current state information and delayed state information. This functional only needs to be positive definite at the sampling instants. Sufficient criteria and the controller design method are provided to ensure the exponential synchronization of CDNs with input saturation under the influence of transmission delay, as well as the estimation of the basin of attraction. Additionally, an optimization algorithm for enlarging the region of attraction is proposed. Finally, a numerical example is presented to verify the effectiveness of the conclusion.

## 1. Introduction

In recent years, complex dynamical networks (CDNs) have garnered significant research attention due to their extensive applicability in various practical systems, such as biological networks, power grids, and communication networks [1,2,3]. The synchronization of these networks, a vital dynamic property, has become a focal point of study due to its relevance in fields like image recognition, secure communication, and information processing [4,5]. Numerous research findings on the synchronization problem of CDNs have been published [6,7,8,9,10,11]. The cluster synchronization for a class of CDNs with mismatched parameters was studied in [12]. Based on the free-weighting matrix approach and a fresh Lyapunov functional, the finite-time synchronization issue of CDNs was investigated in [13]. Considering the controller gains with uncertainties, the non-fragile synchronization control issues of CDNs were studied in [14,15,16].

As a crucial method in networked control, sampled-data control (SDC) only requires storing system information at sampling times, significantly reducing information transmission and thereby saving limited network bandwidth, making it more economical and feasible. Some significant results on sampled-data synchronization control for CDNs have been achieved [17,18,19,20,21,22]. The global $H_{\infty }$ pinning synchronization problem for CDNs via aperiodic SDC was explored in [18], using the input delay method and free-weighting matrix techniques. In [19], the condition for sampled-data synchronization of reaction–diffusion CDNs with semi-Markovian jumping topologies was presented. In [20], the synchronization problem of switched CDNs in a finite time sense was studied with an average dwell-time switching signal. For a class of fractional and impulsive complex networks with switching topologies at impulsive instants, a sampled-data-based event-triggered synchronization control was designed in [21]. By decomposing complex-valued dynamic systems into two real-valued systems, the synchronization of delayed complex-valued chaotic Lur’e systems was studied based on SDC in [22]. The sampling period is a key factor in SDC systems; a longer period can more effectively conserve network resources. Consequently, many researchers have been devoted to reducing conservatism in this aspect [23,24,25].

Actuator saturation frequently occurs in practical systems. Addressing input saturation is critical in controller design, as neglecting this issue can result in reduced system performance and potential instability [26]. Input saturation confines the system to achieve only local stability, with the precise determination of the basin of attraction (BoA) ensuring that stability is limited and challenging to ascertain accurately. Some scholars are dedicated to CDNs and have already carried out some meaningful work. Some synchronization criteria for continuous and discrete systems were presented in [27,28]. In [29], based on the aperiodic SDC method, the synchronization problem was studied and an optimal algorithm for the estimation of BoA was given. By designing parameter-dependent loop-based Lyapunov functionals, the design methods of sampled-data synchronization controller for polytopic uncertain switched CDNs were presented in [30]. Based on the aperiodic intermittent SDC method for complex-valued stochastic systems, the mean-squared exponential synchronization problem was studied in [31]. However, the effect of transmission delay is not considered in these results. Due to the delay in sending the updated signal from the sampler to the controller and then to the zero-order holder, it is necessary to design a memory sampling controller that accounts for transmission delay. Although some significant results for CDNs have considered transmission delay μ, only the information at time $t_{\kappa }$ has been taken into account. Furthermore, considering the input saturation, the question of how to design a memory sampling controller that incorporates comprehensive information about the accessible characteristics of the real sampling pattern and provides an estimation of the BoA is a significant problem to be solved.

According to the above analysis, the main purpose of this paper is to solve the exponential synchronization issue of CDNs and provide the design method of a memory sampled-data controller. The following is a brief description of the main work and innovation:(1)Considering the influence of input saturation and transmission delay, an exponential sampled-data synchronization control scheme for CDNs is proposed.(2)A new two-sided looped-functional containing complete sampling interval information is constructed, and the delay states $z(t_{\kappa }-\mu )$ and $z(t_{\kappa +1}-\mu )$ are also taken into account.(3)Based on the memory SDC strategy and the constructed functional, the estimation of BoA and the corresponding optimization algorithm are given.

The remainder of this paper is organized as follows. In Section 2, the model description and some necessary preliminary knowledge are briefly outlined. The main theoretical conclusion is given in Section 3. In Section 4, a numerical simulation is chosen to validate the effectiveness of the proposed method. Section 5 is the conclusion.

**Notation** **1.**
*$S_{j}=\left\{1,2,\ldots ,j\right\}$. diag{⋯} and col{⋯} represent the block diagonal matrix and column vector or matrix, respectively. $\mathrm{Sym}\left\{V\right\}=V+V^{T}$.*


## 2. Model Description and Preliminaries

Consider the following CDNs with actuator saturation:(1)$$\begin{array}{cc} & {\dot{x}}_{i}\left(t\right)=-f\left(x_{i}\left(t\right)\right)+c\sum_{j=1}^{N}a_{ij}Bx_{i}\left(t\right)+\mathrm{sat}\left(u_{i}\left(t\right)\right) \end{array}$$
for $i\in S_{N}$, where *N* is the number of coupled nodes; $x_{i}\left(t\right)\in R^{n}$ denotes the state vector; *c* is the coupling strength; $A={\left[a_{ij}\right]}_{N\times N}$ and $B\in R^{n\times n}$ are the external and internal coupling matrices, respectively; Matrix *A* satisfies the zero-row-sum condition, where $a_{ij}>0$ if nodes *i* and *j* are connected, and $a_{ij}=0$ otherwise. $f\left(x_{i}\left(t\right)\right):R^{n}\to R^{n}$ is a nonlinear function; $u_{i}\left(t\right)$ represents the control input; the saturation function $\mathrm{sat}\left(u_{i}\left(t\right)\right)={\left[\mathrm{sat}\left(u_{i1}\left(t\right)\right),\mathrm{sat}\left(u_{i2}\left(t\right)\right),\ldots ,\mathrm{sat}\left(u_{in}\left(t\right)\right)\right]}^{T}$ is defined as:$$\begin{array}{c} \mathrm{sat}\left(u_{il}\left(t\right)\right)=\mathrm{sgn}\left(u_{il}\left(t\right)\right)\min\{u_{0l},|u_{l}\left(t\right)\left|\right\},\;l\in S_{n} \end{array}$$
where $u_{0l}>0$ represents the saturation threshold.

Consider the isolated node as follows:(2)$$\begin{array}{c} \dot{y}\left(t\right)=f\left(y\left(t\right)\right),\;y\left(t\right)\in R^{n}. \end{array}$$

Denote the error vectors $z_{i}\left(t\right)=x_{i}\left(t\right)-y\left(t\right)$, and $f\left(z_{i}\left(t\right)\right)=f\left(x_{i}\left(t\right)\right)-f\left(y\left(t\right)\right)$, the error system can be expressed as
(3)$$\begin{array}{cc} {\dot{z}}_{i}\left(t\right)= & -f\left(z_{i}\left(t\right)\right)+c\sum_{j=1}^{N}a_{ij}Bz_{i}\left(t\right)+\mathrm{sat}\left(u_{i}\left(t\right)\right) \end{array}$$

The control signal is transmitted to the plant via the zero-order Hold, utilizing the following hold-time sequence: $0=t_{0}<t_{1}\cdots <\lim_{\kappa \to \infty }t_{\kappa }=+\infty$. The sampling interval $t_{\kappa +1}-t_{\kappa }=h_{\kappa }\in \left(\overset{ˇ}{h},\hat{h}\right]$, ∀κ≥0, is the aperiodic sampling interval, which falls within the range $\left(\overset{ˇ}{h},\hat{h}\right]$.

Consider the SDC input $u_{i}\left(t\right)$ with transmission delay μ as follows:(4)$$\begin{array}{cc} & u_{i}\left(t\right)=K_{1i}z_{i}\left(t_{\kappa }\right)+K_{2i}z_{i}\left(t_{\kappa }-\mu \right),\;t\in \left[t_{\kappa },t_{\kappa +1}\right) \end{array}$$
where $K_{1i}$ and $K_{2i}$ are the controller gain matrices to be designed with appropriate dimensions.

Define the following dead-zone nonlinear function:(5)$$\begin{array}{cc} & \Psi \left(u_{i}\left(t\right)\right)≜\Psi_{\kappa }^{i}=u_{i}\left(t\right)-\mathrm{sat}\left(u_{i}\left(t\right)\right). \end{array}$$

Substituting (5) into (1), the closed-loop system can be given as
(6)$$\begin{array}{cc} \dot{z}\left(t\right)= & F\left(z\left(t\right)\right)+c\left(A⊗B\right)z\left(t\right)+K\eta_{1}\left(t_{\kappa }\right)-\Psi_{\kappa } \end{array}$$
where $z\left(t\right)={\left[z_{1}^{T}\left(t\right),z_{2}^{T}\left(t\right),\ldots ,z_{N}^{T}\left(t\right)\right]}^{T}$, $K=[K_{1},K_{2}]$, $K_{1}=\mathrm{diag}\left\{K_{11},K_{12},\ldots ,K_{1N}\right\}$, $K_{2}=\mathrm{diag}\left\{K_{21},K_{22},\ldots ,K_{2N}\right\}$, $\eta_{1}\left(t_{\kappa }\right)={\left[z^{T}\left(t_{\kappa }\right),z^{T}\left(t_{\kappa }-\mu \right)\right]}^{T}$, $\Psi_{\kappa }={\left[\Psi_{\kappa }^{1T},\Psi_{\kappa }^{2T},\ldots ,\Psi_{\kappa }^{NT}\right]}^{T}$, $F\left(z\left(t\right)\right)={\left[F^{T}\left(z_{1}\left(t\right)\right),F^{T}\left(z_{2}\left(t\right)\right),\ldots ,F^{T}\left(z_{N}\left(t\right)\right)\right]}^{T}$.

The following hypotheses and lemmas are given to facilitate the subsequent theoretical analysis.

**Assumption** **1.**
*For any $r,s\in R^{n}$, $f_{rs}=f\left(r\right)-f\left(s\right)$, constant matrices $L_{1}$ and $L_{2}$, $f_{rs}$ satisfies*

(7)
$$\begin{array}{c} {\left[f_{rs}-L_{1}\left(r-s\right)\right]}^{T}\left[f_{rs}-L_{2}\left(r-s\right)\right]\leq 0. \end{array}$$



**Lemma** **1**([32])**.**
*Let $s\left(\cdot \right):\left[p,q\right]\to R^{n}$ be a differentiable function, and V>0 be any matrix. Then the following inequality holds:*
$$\begin{array}{c} \left(p-q\right)\int_{q}^{p}{\dot{s}}^{T}\left(s\right)V\dot{s}\left(s\right)ds\geq S_{1}^{T}VS_{1}+3S_{2}^{T}VS_{2} \end{array}$$*where $S_{1}=s\left(p\right)-s\left(q\right)$, $S_{2}=s\left(p\right)+s\left(q\right)-\frac{2}{p-q}\int_{q}^{p}s\left(s\right)ds$.*

**Lemma** **2**([33])**.**
*For a scalar 0<b<1, $U,U^{\prime }\in R^{n}$, positive definite matrices ${ћ}_{1},{ћ}_{2}\in R^{n\times n}$, and any matrices $Y,Y^{\prime }\in R^{n\times n}$, the following inequality is satisfied:*
$$\begin{array}{cc} & \frac{1}{\pi }U^{T}{ћ}_{1}U+\frac{1}{1-\pi }U^{\prime T}{ћ}_{2}U^{\prime }\\ & \geq U^{T}\left[{ћ}_{1}+\left(1-\pi \right)\left({ћ}_{1}-Y{ћ}_{2}^{-1}Y^{T}\right)\right]U+2U^{T}[\pi Y\\ & +\left(1-\pi \right)Y^{\prime }]{ℵ}_{2}+U^{\prime T}\left[{ћ}_{2}+\pi \left({ћ}_{2}-Y^{\prime T}{ћ}_{1}^{-1}Y^{\prime T}\right)\right]U^{\prime }. \end{array}$$

**Lemma** **3**([23])**.**
*For a positive definite matrix A and a nonsingular matrix B, let $A^{\prime }=B^{T}AB$, if the matrix inequality*
$$\begin{array}{c} \left[\begin{array}{cc} ℘ & I\\ I & B+B^{T}-A^{\prime } \end{array}\right]>0 \end{array}$$*holds, then A<℘.*

## 3. Main Result

In this section, the design method for the sampling controller will be presented, and the criteria to ensure local stability of the error system (6) will be proposed. For convenience, the following symbols are defined:$$\begin{array}{cc} & H=t-t_{\kappa },\;H^{\prime }=t_{\kappa +1}-t,r_{0}=0_{Nn\times 13Nn},\\ & r_{†}=\left[\begin{array}{ccc} 0_{Nn\times (†-1)Nn} & I_{Nn} & 0_{Nn\times (13-†)Nn} \end{array}\right],\;†\in S_{13}\\ & \eta_{1}\left(t\right)=\mathrm{col}\left\{z\left(t\right),z\left(t-\mu \right)\right\},\eta_{2}\left(t\right)=col\left\{z\left(t\right),\dot{z}\left(t\right)\right\}\\ & \chi_{1}\left(t\right)=\mathrm{col}\left\{z\left(t\right)-z\left(t_{\kappa }\right),z\left(t-\mu \right)-z\left(t_{k}-\mu \right)\right\}\\ & \chi_{2}\left(t\right)=\mathrm{col}\left\{z\left(t_{\kappa +1}\right)-z\left(t\right),z\left(t_{k+1}-\mu \right)-z\left(t-\mu \right)\right\}\\ & \chi_{3}\left(t\right)=\mathrm{col}\left\{H^{\prime }\chi_{1}\left(t\right),H\chi_{2}\left(t\right)\right\},\chi_{4}\left(t\right)=\mathrm{col}\left\{\chi_{1}\left(t\right),\chi_{2}\left(t\right)\right)\}\\ & \chi_{5}\left(t\right)=\mathrm{col}\left\{z\left(t_{\kappa }\right),z\left(t_{\kappa +1}\right),z\left(t_{\kappa }-\mu \right),z\left(t_{\kappa +1}-\mu \right)\right\},\\ & \chi_{6}\left(t\right)=\mathrm{col}\left\{\chi_{5}\left(t\right),v_{1}\left(t\right),v_{2}\left(t\right)\right\}\\ & N\left(t\right)=\mathrm{col}\{\eta_{2}\left(t\right),\eta_{2}\left(t-\mu \right),\chi_{5}\left(t\right),v_{1}\left(t\right),v_{2}\left(t\right),v_{3}\left(t\right),F\left(z\left(t\right)\right),\Psi_{k}\}\\ & v_{1}\left(t\right)=\frac{1}{H}\int_{t_{\kappa }}^{t}z\left(s\right)ds,\;v_{2}\left(t\right)=\frac{1}{H^{\prime }}\int_{t}^{t_{\kappa +1}}z\left(s\right)ds,\;v_{3}\left(t\right)=\frac{1}{\mu }\int_{t-\mu }^{t}z\left(s\right)ds,\\ & L_{1}=0.5{\left(I⊗L_{1}\right)}^{T}\left(I⊗L_{2}\right)+0.5{\left(I⊗L_{2}\right)}^{T}\left(I⊗L_{1}\right),\\ & L_{2}=-0.5{\left(I⊗L_{1}\right)}^{T}-0.5{\left(I⊗L_{2}\right)}^{T},L=\left[\begin{array}{cc} L_{1} & L_{2}\\ ★ & I \end{array}\right]. \end{array}$$

**Theorem** **1.**
*For given scalars $\epsilon_{1}>0$, h>0, if there exist positive matrices $P\in R^{2nN\times 2nN}$, $O_{\rho }\in R^{nN\times nN}$, $R_{1}\in R^{2nN\times 2nN}$, and $R_{2}\in R^{nN\times nN}$, symmetric matrix $Y\in R^{6nN\times 6nN}$, arbitrary matrices $S_{\rho }\in R^{4nN\times 4nN}$, $X\in R^{2nN\times 2nN}$, $Y_{1}\in R^{2nN\times 2nN}$, $Y_{2}\in R^{2nN\times 2nN}$, $G_{\psi }\in R^{nN\times nN}$, $K_{j}\in R^{nN\times nN}$ and $J\in R^{nN\times 2nN}$, $\psi \in S_{2}$, and diagonal positive matrix $\Lambda_{1}\in R^{nN\times nN}$ such that*

(8)
$$\begin{array}{cc} & \Omega^{1}=\left[\begin{array}{cc} E^{1}+h_{k}E^{2} & \Pi_{1}^{T}Y_{1}\\ ★ & -O_{2}^{\prime } \end{array}\right]<0 \end{array}$$


(9)
$$\begin{array}{cc} & \Omega^{2}=\left[\begin{array}{cc} E^{1}+h_{k}E^{3} & \Pi_{2}^{T}Y_{2}^{T}\\ ★ & -O_{1}^{\prime } \end{array}\right]<0 \end{array}$$


(10)
$$\begin{array}{c} \left[\begin{array}{cc} P & {\left(K_{\left(j\right)}-J_{\left(j\right)}\right)}^{T}\\ ★ & U_{0j}^{2} \end{array}\right]>0 \end{array}$$

*where*

$$\begin{array}{cc} E^{1}=\; & \mathrm{sym}\left\{{\Phi_{1}^{1}}^{T}P\Phi_{1}^{2}+\Omega_{3}^{5T}\left(S_{1}\Omega_{4}^{1}+S_{2}\Omega_{5}\right)+\Omega_{1}^{2T}X\Omega_{2}^{1}+\Omega_{1}^{1T}X\Omega_{2}^{2}\right\}-\Pi_{1}^{T}O_{1}^{\prime }\Pi_{1}\\ & -\Pi_{2}^{T}O_{2}^{\prime }\Pi_{2}+{\Phi_{2}^{1}}^{T}R_{1}\Phi_{2}^{1}-{\Phi_{2}^{2}}^{T}R_{1}\Phi_{2}^{2}+\mathrm{sym}\left\{r_{13}^{T}\Lambda_{1}J\Phi_{1}^{3}-r_{13}^{T}\Lambda_{1}r_{13}\right\}\\ & +\mu^{2}r_{2}^{T}R_{2}r_{2}-{\Phi_{0}^{1}}^{T}R_{2}\Phi_{0}^{1}-3{\Phi_{0}^{2}}^{T}R_{2}\Phi_{0}^{2}-\epsilon_{1}E^{T}LE+\Gamma \\ E^{2}=\; & \mathrm{sym}\left\{\Omega_{3}^{1T}S_{1}\Omega_{4}^{2}+\Omega_{3}^{3T}\left(S_{1}\Omega_{4}^{1}+S_{2}\Omega_{5}\right)+\Omega_{6}^{2T}Y\Omega_{6}^{1}-\frac{1}{h_{\kappa }}\Pi_{1}^{T}Y_{2}\Pi_{2}\right\}\\ & +\Omega_{6}^{1T}Y\Omega_{6}^{1}+h_{\kappa }r_{2}^{T}O_{1}r_{2}-\frac{1}{h_{\kappa }}\Pi_{1}^{T}O_{1}^{\prime }\Pi_{1}\\ E^{3}=\; & \mathrm{sym}\left\{\Omega_{3}^{2T}S_{1}\Omega_{4}^{2}+\Omega_{3}^{4T}\left(S_{1}\Omega_{4}^{1}+S_{2}\Omega_{5}\right)+\Omega_{6}^{3T}Y\Omega_{6}^{1}-\frac{1}{h_{\kappa }}\Pi_{1}^{T}Y_{1}\Pi_{2}\right\}\\ & -\Omega_{6}^{1T}Y\Omega_{6}^{1}+h_{\kappa }r_{2}^{T}O_{2}r_{2}-\frac{1}{h_{\kappa }}\Pi_{2}^{T}O_{2}^{\prime }\Pi_{2}\\ \Gamma_{0}=\; & \rho_{1}r_{1}^{T}G^{T}+\rho_{2}r_{2}^{T}G^{T},E=col\left\{r_{1},r_{12}\right\}\\ \Gamma =\; & \mathrm{sym}\{\Gamma_{0}\left(-r_{2}+r_{12}+C\left(A⊗B\right)r_{1}+K\Phi_{1}^{3}-r_{13}\right)\}\\ \Pi_{1}=\; & col\left\{r_{1}-r_{5},r_{1}+r_{5}-2r_{9}\right\},\;\Omega_{1}^{1}=\left\{r_{1}-r_{5},r_{3}-r_{7}\right\},\;\Omega_{1}^{2}=\left\{r_{2},r_{4}\right\}\\ \Pi_{2}=\; & col\left\{r_{6}-r_{1},r_{1}+r_{6}-2r_{10}\right\},\Omega_{2}^{1}=\left\{r_{6}-r_{1},r_{8}-r_{3}\right\},\;\Omega_{2}^{2}=-\Omega_{1}^{2}\\ \Omega_{3}^{1}=\; & col\left\{\Omega_{1}^{1},r_{0},r_{0}\right\},\Omega_{3}^{2}=col\left\{r_{0},r_{0},\Omega_{2}^{1}\right\},\Omega_{3}^{3}=col\left\{\Omega_{1}^{2},r_{0},r_{0}\right\},\Omega_{3}^{4}=col\left\{r_{0},r_{0},\Omega_{2}^{2}\right\}\\ \Omega_{3}^{5}=\; & col\left\{-\Omega_{1}^{1},\Omega_{2}^{1}\right\},\Omega_{4}^{1}=col\left\{\Omega_{1}^{1},\Omega_{2}^{1}\right\},\Omega_{4}^{2}=col\left\{\Omega_{1}^{2},\Omega_{2}^{2}\right\}\Omega_{5}=col\left\{r_{5},r_{6},r_{7},r_{8}\right\},\\ \Omega_{6}^{1}=\; & col\left\{\Omega_{5},r_{9},r_{10}\right\},\Phi_{0}^{1}=r_{1}-r_{3},\Omega_{6}^{2}=col\left\{r_{0},r_{0},r_{0},r_{0},r_{1}-r_{9},r_{0}\right\},\\ \Omega_{6}^{3}=\; & col\left\{r_{0},r_{0},r_{0},r_{0},r_{0},r_{10}-r_{1}\right\},\Phi_{0}^{2}=r_{1}+r_{3}-2r_{11},\Phi_{1}^{1}=\left\{r_{1},r_{3}\right\}\\ \Phi_{2}^{1}=\; & \left\{r_{1},r_{2}\right\},\;\Phi_{2}^{2}=\left\{r_{3},r_{4}\right\},\;\Phi_{1}^{3}=\left\{r_{5},r_{7}\right\},\Phi_{1}^{2}=\left\{r_{2},r_{4}\right\} \end{array}$$

*then the error system (6) is stable for any initial condition $z\left(t_{0}\right)\in E\left(P\right)=\left\{z\left(t_{0}\right)\in R^{nN}|{♮}_{2}\underset{-r\leq s\leq 0}{\sup}\{∥z\left(s\right){∥}^{2},∥\dot{z}\left(s\right){∥}^{2}\}\leq 1\right\}$.*


**Proof.** Consider the following two-sided looped-functional:
(11)$$\begin{array}{c} V\left(t\right)=V_{1}\left(t\right)+V_{2}\left(t\right)+V_{3}\left(t\right)+V_{4}\left(t\right) \end{array}$$
where
$$\begin{array}{cc} V_{1}\left(t\right) & =\eta_{1}^{T}\left(t\right)P\eta_{1}\left(t\right)\\ V_{2}\left(t\right) & =2\chi_{3}^{T}\left(t\right)\left(S_{1}\chi_{4}\left(t\right)+S_{2}\chi_{5}\left(t\right)\right)+2\chi_{1}^{T}\left(t\right)X\chi_{2}\left(t\right)\\ V_{3}\left(t\right) & =HH^{\prime }\chi_{6}^{T}\left(t\right)Y\chi_{6}\left(t\right)\\ V_{4}\left(t\right) & =H^{\prime }h_{\kappa }\int_{t_{\kappa }}^{t}{\dot{z}}^{T}\left(s\right)O_{1}\dot{z}\left(s\right)ds-Hh_{\kappa }\int_{t}^{t_{\kappa +1}}{\dot{z}}^{T}\left(s\right)O_{2}\dot{z}\left(s\right)ds\\ V_{5}\left(t\right) & =\int_{t-\mu }^{t}\eta_{2}^{T}\left(s\right)R_{1}\eta_{2}\left(s\right)ds+\mu \int_{-\mu }^{0}\int_{t+\varphi }^{t}{\dot{z}}^{T}\left(s\right)R_{2}\dot{z}\left(s\right)dsd\varphi . \end{array}$$Calculating the time derivative of V(t) yields
(12)$$\begin{array}{cc} {\dot{V}}_{1}\left(t\right)=\; & {\dot{\eta }}_{1}{\left(t\right)}^{T}P\eta_{1}\left(t\right)+\eta_{1}^{T}\left(t\right)P{\dot{\eta }}_{1}\left(t\right) \end{array}$$
(13)$$\begin{array}{cc} {\dot{V}}_{2}\left(t\right)=\; & 2{\dot{\chi }}_{1}^{T}\left(t\right)X\chi_{2}\left(t\right)+2\chi_{1}^{T}\left(t\right)X{\dot{\chi }}_{2}\left(t\right)+2{\dot{\chi }}_{3}^{T}\left(t\right)\left(S_{1}\chi_{4}\left(t\right)+S_{2}\chi_{5}\left(t\right)\right)\\ & +2{\dot{\chi }}_{1}^{T}\left(t\right)X\chi_{2}\left(t\right)+2\chi_{1}^{T}\left(t\right)X{\dot{\chi }}_{2}\left(t\right) \end{array}$$
(14)$$\begin{array}{cc} {\dot{V}}_{3}\left(t\right)=\; & 2HH^{\prime }{\dot{\chi }}_{6}^{T}\left(t\right)Y\chi_{6}\left(t\right)+H^{\prime }\chi_{6}^{T}\left(t\right)Y\chi_{6}\left(t\right)-H\chi_{6}^{T}\left(t\right)Y\chi_{6}\left(t\right) \end{array}$$
(15)$$\begin{array}{cc} {\dot{V}}_{4}\left(t\right)=\; & H^{\prime }h_{\kappa }{\dot{z}}^{T}\left(t\right)O_{1}\dot{z}\left(t\right)+Hh_{\kappa }{\dot{z}}^{T}\left(t\right)O_{2}\dot{z}\left(t\right)\\ & -h_{\kappa }\int_{t_{\kappa }}^{t}{\dot{z}}^{T}\left(s\right)O_{1}\dot{z}\left(s\right)ds-h_{\kappa }\int_{t}^{t_{\kappa +1}}{\dot{z}}^{T}\left(s\right)O_{2}\dot{z}\left(s\right)ds \end{array}$$
(16)$$\begin{array}{cc} {\dot{V}}_{5}\left(t\right)=\; & \eta_{2}^{T}\left(t\right)R_{1}\eta_{2}\left(t\right)-\eta_{2}^{T}\left(t-\mu \right)R_{1}\eta_{2}\left(t-\mu \right)\\ & +\mu^{2}{\dot{z}}^{T}\left(t\right)R_{2}\dot{z}\left(t\right)-\mu \int_{t-\mu }^{t}{\dot{z}}^{T}\left(s\right)R_{2}\dot{z}\left(s\right)ds \end{array}$$By applying Lemma 1, we have
(17)$$\begin{array}{cc} & \mu \int_{t-\mu }^{t}{\dot{z}}^{T}\left(s\right)R_{2}\dot{z}\left(s\right)ds\\ & \geq N^{T}\left(t\right)\left[\Phi_{0}^{1T}R_{2}\Phi_{0}^{1}+3\Phi_{0}^{2T}R_{2}\Phi_{0}^{2})\right]N\left(t\right). \end{array}$$From Lemmas 1 and 2, it can be inferred that
(18)$$\begin{array}{cc} & -h_{\kappa }\int_{t_{\kappa }}^{t}e^{2as}{\dot{z}}^{T}\left(s\right)O_{1}\dot{z}\left(s\right)ds-h_{\kappa }\int_{t}^{t_{\kappa +1}}{\dot{z}}^{T}\left(s\right)O_{2}\dot{z}\left(s\right)ds\\ & \;\;\leq -\left[\frac{h_{\kappa }}{H}Q_{1}^{T}O_{1}Q_{1}+\frac{h_{\kappa }}{H^{\prime }}Q_{2}^{T}O_{2}Q_{2}\right]\\ & \;\;\leq -N^{T}\left(t\right)[\Pi_{1}^{T}\left(O_{1}+\frac{H^{\prime }}{h_{\kappa }}\left(O_{1}-Y_{1}O_{2}^{-1}Y_{1}^{T}\right)\right)\Pi_{1}\\ & \;\;\;\;\;\;\;\;\;\;\;\;+\Pi_{2}^{T}\left(O_{2}+\frac{H}{h_{\kappa }}\left(O_{2}-Y_{2}^{T}O_{1}^{-1}Y_{2}\right)\right)\Pi_{2}\\ & \;\;\;\;\;\;\;\;\;\;\;\;+\mathrm{sym}\left\{\Pi_{1}^{T}\left(\frac{H}{h_{\kappa }}Y_{1}+\frac{H^{\prime }}{h_{\kappa }}Y_{2}\right)\Pi_{2}\right\}]N\left(t\right) \end{array}$$
where
$$\begin{array}{cc} Q_{1} & =col\{z\left(t\right)-z\left(t_{\kappa }\right),z\left(t\right)+z\left(t_{\kappa }\right)-2v_{1}\left(t\right)\}\\ Q_{2} & =col\{z\left(t_{\kappa +1}\right)-z\left(t\right),z\left(t_{\kappa +1}\right)+z\left(t\right)-2v_{2}\left(t\right)\}\\ O_{1} & =\mathrm{diag}\left\{O_{1},3O_{1}\right\},\;O_{2}=\mathrm{diag}\left\{O_{2},3O_{2}\right\}. \end{array}$$With regard to AS, defining a polyhedral set
(19)$$\begin{array}{c} H=\left\{Z\in R^{2nN}|\left|\left(K_{\left(j\right)}-\Upsilon_{\left(j\right)}\right)Z\right|\leq u_{0l},j\in S_{nN}\right\} \end{array}$$
where $J\in R^{nN\times 2nN}$, $K_{\left(j\right)}$ and $J_{\left(j\right)}$ denote the *j*th row of the matrices K and *J*, respectively. Then, according to (10), the following sector condition holds:
(20)$$\begin{array}{c} 0\leq \mathrm{sym}\{\Psi_{\kappa }^{T}\Xi \Upsilon \eta_{1}\left(t_{\kappa }\right)-\Psi_{\kappa }^{T}\Xi \Psi_{\kappa }\} \end{array}$$
for $\eta_{1}\left(t_{\kappa }\right)\in ⋄$ and any diagonal matrix Ξ>0.According to the dynamic Equation (6), for any matrices $G_{1}$ and $G_{2}$, it yields that
(21)$$\begin{array}{cc} (G_{1}z\left(t\right) & +G_{2}\dot{z}\left(t\right){)}^{T}(-\dot{z}\left(t\right)+F\left(z\left(t\right)\right)\\ & +c\left(A⊗B\right)z\left(t\right)+K\eta_{1}\left(t_{\kappa }\right)-\Psi_{\kappa })=0. \end{array}$$From the Assumption 1, we have
(22)$$\begin{array}{cc} & \epsilon_{1}N^{T}\left(t\right)E^{T}LEN\left(t\right)\leq 0. \end{array}$$Combining the above Formulas (11)–(22) results in
(23)$$\begin{array}{cc} \dot{V}\left(t\right) & \leq N^{T}\left(t\right)(\frac{H^{\prime }}{h_{\kappa }}\left(E^{1}+h_{\kappa }E^{2}+\Pi_{1}^{T}Y_{1}O_{2}^{-1}Y_{1}^{T}\Pi_{1}\right)\\ & \;\;\;+\frac{H}{h_{\kappa }}\left(E^{1}+h_{\kappa }E^{3}+\Pi_{2}^{T}Y_{2}O_{1}^{-1}Y_{2}^{T}\Pi_{2}\right))N\left(t\right)\\ & ≜N^{T}\left(t\right)\Omega_{0}N\left(t\right). \end{array}$$Based on the Schur complement, it follows from the LMIs (8) and (9) that $\Omega_{0}<0$. Consequently, we have
(24)$$\begin{array}{c} \dot{V}\left(t\right)<0,\;t\in \left[t_{\kappa },t_{\kappa +1}\right), \end{array}$$
which indicates
(25)$$\begin{array}{cc} & V\left(t\right)\leq V\left(t_{\kappa }\right)\leq V\left(t_{\kappa -1}\right)\leq \ldots \leq V\left(0\right) \end{array}$$It follows from (20) that there exists a constant ♮ such that
(26)$$\begin{array}{cc} & \dot{V}\left(t\right)\leq -\varrho_{1}{∥z\left(t_{0}\right)∥}^{2}. \end{array}$$Moreover, from (6), we have
(27)$$\begin{array}{cc} V(0)=\; & \eta_{1}^{T}\left(0\right)P\eta_{1}\left(0\right)+\int_{-\mu }^{0}\eta_{2}^{T}\left(s\right)R_{1}\eta_{2}\left(s\right)ds+\mu \int_{-\mu }^{0}\int_{\varphi }^{0}{\dot{z}}^{T}\left(s\right)R_{2}\dot{z}\left(s\right)dsd\varphi \\ \leq \; & \lambda_{\max}\left(P\right)∥\eta_{1}{\left(0\right)∥}^{2}+2\eta \lambda_{\max}\left(R_{1}\right)\underset{-r\leq s\leq 0}{\sup}\{∥z\left(s\right){∥}^{2},∥\dot{z}\left(s\right){∥}^{2}\}\\ & +\eta^{3}\lambda_{\max}\left(R_{2}\right)\underset{-r\leq s\leq 0}{\sup}\{∥\dot{z}\left(s\right){∥}^{2}\}\\ \leq \; & (2\lambda_{\max}\left(P\right)+2\eta \lambda_{\max}\left(R_{1}\right)\\ & +\eta^{3}\lambda_{\max}\left(R_{2}\right))\underset{-r\leq s\leq 0}{\sup}\{∥z\left(s\right){∥}^{2},∥\dot{z}\left(s\right){∥}^{2}\} \end{array}$$Define $\tilde{V}\left(t\right)=e^{2at}V\left(t\right)$, where *a* is a positive constant. It can be deduced from (26) and (27) that
(28)$$\begin{array}{cc} \dot{\tilde{V}}\left(t\right)=\; & e^{2at}\dot{V}\left(t\right)+2ae^{2at}V\left(t\right)\\ \leq \; & -e^{2at}{♮}_{1}{∥z\left(t_{0}\right)∥}^{2}+2ae^{2at}V\left(0\right)\\ \leq \; & -e^{2at}\left(\varrho_{1}-2a\varrho_{2}\right)\underset{-r\leq s\leq 0}{\sup}{\{∥z\left(s\right)∥}^{2},∥\dot{z}\left(s\right){∥}^{2}\} \end{array}$$
where $\varrho_{2}=\lambda_{\max}\left(P\right)+2\mu \lambda_{\max}\left(R_{1}\right)+\mu^{3}\lambda_{\max}\left(R_{2}\right)$.By choosing $a<\varrho_{1}/\left(2\varrho_{2}\right)$, we obtain
(29)$$\begin{array}{c} \dot{\tilde{V}}\left(t\right)<0 \end{array}$$
which indicates that
(30)$$\begin{array}{c} e^{2at}V\left(t\right)<e^{2at_{\kappa }}V\left(t_{\kappa }\right)<\ldots <V\left(0\right). \end{array}$$For system (6), using the Cauchy inequality, we have
(31)$$\begin{array}{cc} {∥z\left(t\right)∥}^{2} & \leq 6∥z\left(t_{\kappa }\right){∥}^{2}+6\int_{t_{\kappa }}^{t}{∥F\left(z\left(s\right)\right)∥}^{2}ds\\ & +6\int_{t_{\kappa }}^{t}{∥C\left(A⊗B\right)z\left(s\right)∥}^{2}ds+6\int_{t_{\kappa }}^{t}{∥K_{1}z\left(t_{\kappa }\right)∥}^{2}ds\\ & +6\int_{t_{\kappa }}^{t}{∥K_{2}z\left(t_{\kappa }-\eta \right)∥}^{2}ds\\ & \leq 6∥z\left(t_{\kappa }\right){∥}^{2}+6\left(L^{2}+C\left(A⊗B\right)\right)\int_{t_{\kappa }}^{t}{∥z\left(s\right)∥}^{2}ds\\ & +6h_{\kappa }^{2}(∥K_{1}{∥}^{2}∥z\left(t_{\kappa }\right){∥}^{2}+∥K_{2}{∥}^{2}∥z\left(t_{\kappa }-\eta \right){∥}^{2}) \end{array}$$By further applying the Gronwall–Bellman inequality, we derive
(32)$$\begin{array}{cc} {∥z\left(t\right)∥}^{2}\leq \; & 6∥z\left(t_{\kappa }\right){∥}^{2}e^{h_{\kappa }W_{2}}+W_{3}e^{h_{\kappa }W_{2}}{∥z\left(t_{\kappa }\right)∥}^{2}\\ & \;+W_{4}e^{h_{\kappa }W_{2}}{∥z\left(t_{\kappa }-\eta \right)∥}^{2}\\ \leq \; & \frac{6+W_{3}}{\lambda_{\min}\left(P\right)}e^{h_{\kappa }W_{2}}V\left(t_{\kappa }\right)+\frac{W_{4}}{\lambda_{\min}\left(P\right)}e^{h_{\kappa }W_{2}}V\left(t_{\kappa }\right)\\ \leq \; & W^{*}e^{-2at_{\kappa }}{♮}_{2}\underset{-r\leq s\leq 0}{\sup}{\{∥z\left(s\right)∥}^{2},∥\dot{z}\left(s\right){∥}^{2}\} \end{array}$$
where
$$\begin{array}{cc} & W_{2}=6\left(L^{2}+C\left(A⊗B\right)\right),W_{3}=6h_{\kappa }^{2}{∥K_{1}∥}^{2},\\ & W_{4}=6h_{\kappa }^{2}{∥K_{2}∥}^{2},W^{*}=\frac{6+W_{3}+W_{4}}{\lambda_{\min}\left(P\right)}e^{h_{\kappa }W_{2}}. \end{array}$$Therefore, we obtain
(33)$$\begin{array}{cc} & ∥\dot{z}\left(t\right)∥\leq e^{-at}\sqrt{e^{2ah_{\kappa }}W^{*}{♮}_{2}}\underset{-r\leq s\leq 0}{\sup}\{∥z\left(s\right)∥,∥\dot{z}\left(s\right)∥\}. \end{array}$$Thus, system (6) exponentially converges to the origin for any initial conditions $z\left(t_{0}\right)\in E\left(P\right)$. This completes the proof. □

**Theorem** **2.**
*For given scalars $\epsilon_{1}>0$, h>0, if there exist matrices $P\in R^{2nN\times 2nN}>0$, $O_{\rho }\in R^{nN\times nN}>0$, $R_{1}\in R^{2nN\times 2nN}>0$, $R_{2}\in R^{nN\times nN}>0$, symmetric matrix $Y\in R^{6nN\times 6nN}$, arbitrary matrices $S_{\rho }\in R^{4nN\times 4nN}$, $X\in R^{2nN\times 2nN}$, $Y_{1}\in R^{2nN\times 2nN}$, $Y_{2}\in R^{2nN\times 2nN}$, $G_{\psi }\in R^{nN\times nN}$, $K_{j}\in R^{nN\times nN}$, $J\in R^{nN\times 2nN}$, $\psi \in S_{2}$, and diagonal matrix ${\bar{\Lambda }}_{1}\in R^{nN\times nN}>0$ such that*

(34)
$$\begin{array}{cc} & {\bar{\Omega }}^{1}=\left[\begin{array}{cc} E^{1}+h_{k}E^{2} & \Pi_{1}^{T}Y_{1}\\ ★ & -O_{2}^{\prime } \end{array}\right]<0 \end{array}$$


(35)
$$\begin{array}{cc} & {\bar{\Omega }}^{2}=\left[\begin{array}{cc} E^{1}+h_{k}E^{3} & \Pi_{2}^{T}Y_{2}^{T}\\ ★ & -O_{1}^{\prime } \end{array}\right]<0 \end{array}$$


(36)
$$\begin{array}{c} \left[\begin{array}{cc} P & {\left(K_{\left(j\right)}-J_{\left(j\right)}\right)}^{T}\\ ★ & U_{0j}^{2} \end{array}\right]>0 \end{array}$$

*where*

$$\begin{array}{cc} E^{1}=\; & \mathrm{sym}\left\{{\Phi_{1}^{1}}^{T}P\Phi_{1}^{2}+\Omega_{3}^{5T}\left(S_{1}\Omega_{4}^{1}+S_{2}\Omega_{5}\right)+\Omega_{1}^{2T}X\Omega_{2}^{1}+\Omega_{1}^{1T}X\Omega_{2}^{2}\right\}-\Pi_{1}^{T}O_{1}^{\prime }\Pi_{1}\\ & -\Pi_{2}^{T}O_{2}^{\prime }\Pi_{2}+{\Phi_{2}^{1}}^{T}R_{1}\Phi_{2}^{1}-{\Phi_{2}^{2}}^{T}R_{1}\Phi_{2}^{2}+\mathrm{sym}\left\{r_{13}^{T}J\Phi_{1}^{3}-r_{13}^{T}{\bar{\Lambda }}_{1}r_{13}\right\}\\ & +\mu^{2}r_{2}^{T}R_{2}r_{2}-{\Phi_{0}^{1}}^{T}R_{2}\Phi_{0}^{1}-3{\Phi_{0}^{2}}^{T}R_{2}\Phi_{0}^{2}-\epsilon_{1}E^{T}\bar{L}E+\bar{\Gamma }\\ E^{2}=\; & \mathrm{sym}\left\{\Omega_{3}^{1T}S_{1}\Omega_{4}^{2}+\Omega_{3}^{3T}\left(S_{1}\Omega_{4}^{1}+S_{2}\Omega_{5}\right)+\Omega_{6}^{2T}Y\Omega_{6}^{1}-\frac{1}{h_{\kappa }}\Pi_{1}^{T}Y_{2}\Pi_{2}\right\}\\ & +\Omega_{6}^{1T}Y\Omega_{6}^{1}+h_{\kappa }r_{2}^{T}O_{1}r_{2}-\frac{1}{h_{\kappa }}\Pi_{1}^{T}O_{1}^{\prime }\Pi_{1}\\ E^{3}=\; & \mathrm{sym}\left\{\Omega_{3}^{2T}S_{1}\Omega_{4}^{2}+\Omega_{3}^{4T}\left(S_{1}\Omega_{4}^{1}+S_{2}\Omega_{5}\right)+\Omega_{6}^{3T}Y\Omega_{6}^{1}-\frac{1}{h_{\kappa }}\Pi_{1}^{T}Y_{1}\Pi_{2}\right\}\\ & -\Omega_{6}^{1T}Y\Omega_{6}^{1}+h_{\kappa }r_{2}^{T}O_{2}r_{2}-\frac{1}{h_{\kappa }}\Pi_{2}^{T}O_{2}^{\prime }\Pi_{2}\\ {\bar{\Gamma }}_{0}=\; & \rho_{1}r_{1}^{T}+\rho_{2}r_{2}^{T},E=\mathrm{col}\left\{r_{1},r_{12}\right\}\\ \bar{\Gamma }=\; & \mathrm{sym}\{{\bar{\Gamma }}_{0}\left(-G_{1}r_{2}+G_{1}r_{12}+c\left(A⊗B\right)G_{1}r_{1}+K\Phi_{1}^{3}-G_{1}r_{13}\right)\} \end{array}$$

*then the error system (6) is stable for any initial condition $z\left(t_{0}\right)\in E\left(G_{2}^{-T}PG_{2}\right)$. The parameter matrices are given as $K=KG_{2}^{-1}$.*


**Proof.** Let
$$\begin{array}{cc} & G_{1}=\rho_{1}G,G_{2}=\rho_{2}G,\begin{array}{c} G_{\vartheta }=\mathrm{diag}\left\{\underset{︸}{G^{-1},G^{-1},\ldots ,G^{-1}}\right\}\\ \;\;\;\;\;\;\;\;\;\;\;\;\;\vartheta \end{array},\;\vartheta \in S_{13},R_{1}=G_{2}^{T}R_{1}G_{2}R_{2}=G_{1}^{T}R_{2}G_{1}\\ & J=JG_{2},P=G_{2}^{T}PG_{2},\;S_{\rho }=G_{4}^{T}S_{\rho }G_{4},\;Y_{\rho }=G_{2}^{T}Y_{\rho }G_{2}\\ & O_{\rho }=G_{1}^{T}O_{\rho }G_{1},X=G_{2}^{T}XG_{1},Y=G_{4}^{T}YG_{4},\bar{\Lambda }=\Lambda^{-1} \end{array}$$By pre- and post-multiplying (8) and (9) via $\mathrm{diag}\{G_{12}^{T},{\bar{\Lambda }}^{T},G_{2}^{T}\}$ and $\mathrm{diag}\{G_{12},\bar{\Lambda },G_{2}\}$, the condition (34) and (35) can be obtained, and the inequality (36) holds by pre- and post-multiplying (10) via $\mathrm{diag}\{G_{2}^{T},I\}$ and $\mathrm{diag}\{G_{2},I\}$. The controller gain matrices are calculated as $K=KG_{2}^{-1}$. □

**Remark** **1.**
*Given the construction of $V_{1}\left(t\right)$, the traditional quadratic-form invariant set requirements for initial values are no longer applicable. Notably, the functional V(t) reduces to $V_{1}\left(t_{\kappa }\right)+V_{5}\left(t_{\kappa }\right)$ at $t=t_{\kappa }$. Consequently, the estimated BoA is defined as $E\left(P\right)=\left\{z\left(t_{0}\right)\in R^{nN}|{♮}_{2}\underset{-r\leq s\leq 0}{\sup}\{∥z\left(s\right){∥}^{2},∥\dot{z}\left(s\right){∥}^{2}\}\leq 1\right\}$ in this paper. Utilizing the Schur complement and (10), it follows that $\eta_{1}^{T}\left(t_{0}\right){\left(K_{\left(l\right)}-J_{\left(l\right)}\right)}^{T}u_{0l}^{2}\left(K_{\left(l\right)}-J_{\left(l\right)}\right)\eta_{1}\left(t_{0}\right)\leq \eta_{1}^{T}\left(t_{0}\right)P\eta_{1}\left(t_{0}\right)\leq V_{1}\left(t_{0}\right)+V_{5}\left(t_{0}\right)\leq {♮}_{2}\underset{-r\leq s\leq 0}{\sup}\{∥z\left(s\right){∥}^{2},∥\dot{z}\left(s\right){∥}^{2}\}\leq 1$, indicating $\eta_{1}\left(t_{0}\right)\in \delta$. Thus, $\mathrm{sym}\{\Psi_{0}^{T}\Lambda_{1}J\eta_{1}\left(t_{0}\right)-\Psi_{0}^{T}\Lambda_{1}\Psi_{0}\}\geq 0$ holds. From (24), we deduce $\eta_{1}^{T}\left(t_{1}\right){\left(K_{\left(l\right)}-J_{\left(l\right)}\right)}^{T}u_{0l}^{2}\left(K_{\left(l\right)}-J_{\left(l\right)}\right)\eta_{1}\left(t_{1}\right)\leq \eta_{1}^{T}\left(t_{1}\right)P\eta_{1}\left(t_{1}\right)\leq V_{1}\left(t_{1}\right)+V_{5}\left(t_{1}\right)\leq V_{1}\left(t_{0}\right)+V_{5}\left(t_{0}\right)\leq 1$, implying $\eta_{1}\left(t_{1}\right)\in \delta$. By repeating this process, it can be concluded that the sector condition (20) is satisfied.*


Building on these theoretical results, an optimization procedure will be developed to enhance the BoA by determining an allowable initial state region.

For the given *h*, $\epsilon_{1}$ and $u_{oi}$, we can determine the gain matrices K and the maximal estimate of BoA by solving the following optimization problem:$$\begin{array}{cc} & \min\mu \end{array}$$
s.t. (33)–(35), and
(37)$$\begin{array}{cc} & \left[\begin{array}{cc} \mu I & I\\ I & G_{2}+G_{2}^{T}-P \end{array}\right]>0. \end{array}$$

Considering $P=G_{2}^{T}PG_{2}$ and applying Lemma 3 ensures P<μI, which implies $\lambda_{\max}\left(P\right)<\mu$.

**Remark** **2.**
*The matrices $S_{1}$, $S_{2}$, X and Y are not required to be symmetric or positive definite. This relaxes the positivity constraint on the functional V(t), allowing for less conservative results. As a result, the traditional continuous Lyapunov stability theory, which asserts $\dot{V}\left(t\right)<0$ for $t\in [t_{\kappa },t_{\kappa +1})$, no longer implies that $\lim_{t\to \infty }z\left(t\right)=0$. The terms $z\left(t_{\kappa }-\mu \right)-z\left(t-\mu \right)$ and $z\left(t_{\kappa +1}-\mu \right)-z\left(t-\mu \right)$ are included in V(t), with the crucial term z(t−μ) appearing in $V_{1}\left(t\right)$. The stability of SDC systems using looped functionals based on discrete stability theory and inequality techniques was discussed in this paper.*


## 4. Simulation Results

In this section, a numerical example will be presented to illustrate the effectiveness of the theoretical analysis.

Consider system (Equation 1) with n=2 and N=3, the parameter matrices are given [14,34] $A=\left[\begin{array}{ccc} -1 & 0 & 1\\ 0 & -1 & 0\\ 1 & 1 & 2 \end{array}\right]$, $B=\left[\begin{array}{cc} 1 & 0\\ 0 & 1 \end{array}\right]$. and nonlinear function
$$\begin{array}{c} f\left(x_{i}\left(t\right)\right))=\left[\begin{array}{c} -0.5x_{i1}\left(t\right)+\tanh\left(0.2x_{i1}\left(t\right)\right)+0.2x_{i2}\left(t\right)\\ 0.9x_{i2}\left(t\right)-\tanh\left(0.75x_{i2}\left(t\right)\right) \end{array}\right]. \end{array}$$
Then, we can obtain
$$\begin{array}{c} L_{1}=\left[\begin{array}{cc} -0.3 & 0.2\\ 0 & 0.2 \end{array}\right],\;L_{2}=\left[\begin{array}{cc} -0.5 & 0.2\\ 0 & 0.95 \end{array}\right] \end{array}$$

Taking the coupling strength c=0.2, $u_{0il}=1$, and μ=0.1. By solving Theorem 2, we can obtain h=0.5881 and
$$\begin{array}{cc} & K_{11}=\left[\begin{array}{cc} -0.4905 & 0.8800\\ -0.4496 & 0.4575 \end{array}\right],\;K_{12}=\left[\begin{array}{cc} -0.5786 & -0.0464\\ -0.6810 & -0.0883 \end{array}\right]\\ & K_{13}=\left[\begin{array}{cc} -0.5786 & -0.0464\\ -0.6810 & -0.0883 \end{array}\right],\;K_{21}=\left[\begin{array}{cc} -0.4905 & 0.8800\\ -0.4496 & 0.4575 \end{array}\right]\\ & K_{22}=\left[\begin{array}{cc} -0.5786 & -0.0464\\ -0.6810 & -0.0883 \end{array}\right],\;K_{23}=\left[\begin{array}{cc} -0.5786 & -0.0464\\ -0.6810 & -0.0883 \end{array}\right]. \end{array}$$
and, the largest BoA can be calculated as
$$\begin{array}{cc} & E=\left\{z\left(t_{0}\right)\in R^{6}|0.0323\underset{-0.1\leq s\leq 0}{\sup}\{∥z\left(s\right){∥}^{2},∥\dot{z}\left(s\right){∥}^{2}\}\leq 1\right\}. \end{array}$$

The initial conditions are set as $x\left(t_{0}\right)={\left[3,-5;4,-3;-4,-2\right]}^{T}$ and $y\left(0\right)={\left[2,-1\right]}^{T}$. When $u_{i}=0$, the state response of the open-loop error system is shown in Figure 1. The state response of the closed-loop system is shown in Figure 2. It can be seen from Figure 2 that the local stabilization of the error system (6) can be achieved, which indicates that the CDNs (Equation 1) and isolated node (2) are exponentially synchronized under the controller (4). Figure 3 depicts the curves of the control input $u_{i}\left(t\right)$, and the aperiodic sampling interval for the initial state.

## 5. Conclusions

In this paper, based on SDC, the issue of exponential synchronization control for CDNs with input saturation has been studied. Considering the impact of transmission delay μ, a design method for a memory sampled-data controller has been proposed. By constructing an improved two-sided looped functional that includes both current state information and delayed state information, where both $z(t_{\kappa }-\mu )$ and $z(t_{\kappa +1}-\mu )$ have been considered, and combining it with inequality techniques, the criteria for ensuring the local stability of the closed-loop system have been provided. Meanwhile, under the influence of transmission delay, an estimation of the BoA for the initial values has been given, along with an optimization algorithm to enlarge it. Finally, a numerical example has been presented to verify the effectiveness of the results. Due to the susceptibility of complex network environments to external attacks, the question of how to achieve effective anti-attack security control under different types of attacks is a topic worthy of future research.

## Figures and Tables

**Figure 1 entropy-26-00785-f001:**
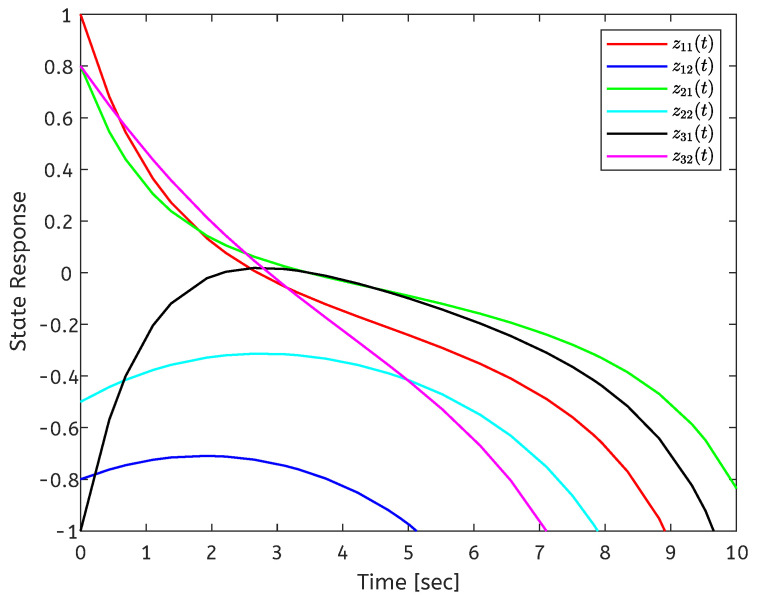
State trajectory of the error system without controller.

**Figure 2 entropy-26-00785-f002:**
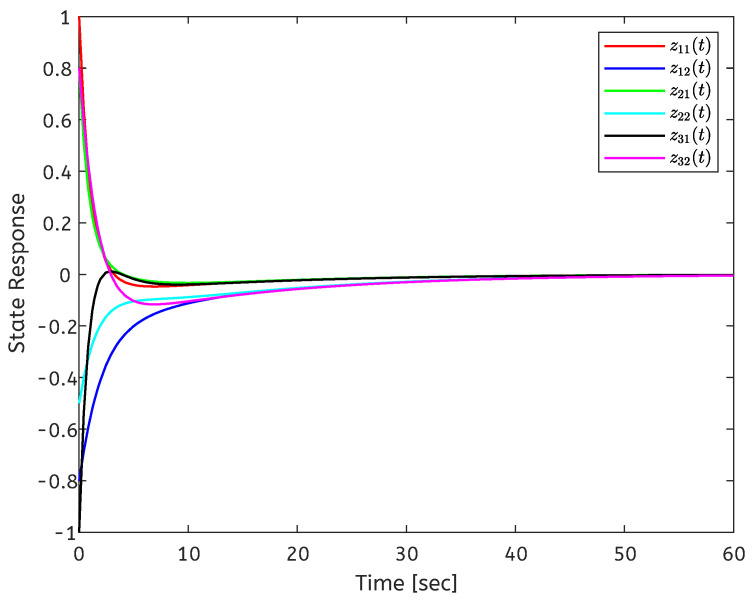
State response of the closed-loop system.

**Figure 3 entropy-26-00785-f003:**
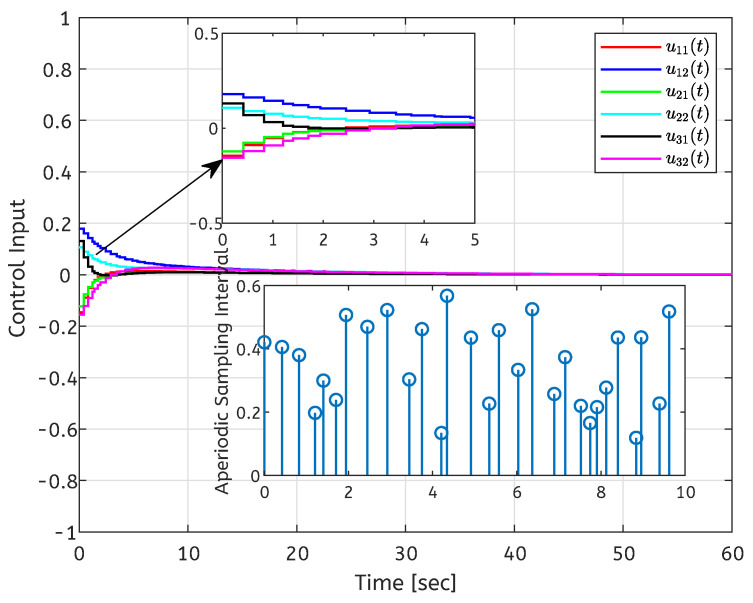
Control inputs and aperiodic sampled-data interval.

## Data Availability

Data are contained within the article.

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
