# Peer review of "Sampled-Data Exponential Synchronization of Complex Dynamical Networks with Saturating Actuators"

_entropy, 2024, doi:10.3390/e26090785_

Round 1

Reviewer 1 Report

Comments and Suggestions for Authors

This paper investigates the exponential synchronization control for CDNs with input saturation. a memory sampled-data controller is designed. Based on the modified two-sided looped functional, both current state information and delayed state information are included. Sufficient criteria and controller design method are provided to ensure the exponential synchronization of CDNs. The estimation of the BoA is given, and an optimization algorithm is proposed.

The overall presentation of the paper is good and the results are correct. Below is a list of comments which the authors might like to take into account when revising the paper:

Comment 1. Some symbols are not clear, for example, $t-\mu$ and $t-\eta$, please check the full text carefully.

Comment 2. Some parts of mathematical derivations are not given in details. I suggest the authors a carefully checking and give all the necessary manipulations for the derived formulas.

Comment 3. The full name and abbreviation need to be consistent.

Comment 4. $\mathcal{Y}_1$ and $ \mathcal{Y}_2$ in Theorem 2 should be given.

Comment 5. The formula reference is incorrect, for example, the part above formula (20).

Comments on the Quality of English Language

Minor revision

Reviewer 2 Report

Comments and Suggestions for Authors

General comments:
-----------------
In the present paper, based on the sampled-data control, the issue of exponential synchronization control with input saturation for complex dynamical networks defined in (1) has been studied. Numerical example was presented to illustrate the theoretical results (page 9).

The main results, summarized as Theorems 1 and 2 (Pages 4 and 8), are very technical in nature with assumptions that are, in my opinion, hard to impossible to verify in concrete cases!

The paper presents some new results and extends the areas of knowledge in control theory, so I recommend it for publication in Entropy after some revisions.

Specific comments:
------------------------

Line 77: Usually the set of symmetric matrices are referred to as Sn, but here it will be something else, but it is not clear what.

Line 108 (Lemma 3): "positive matrix A" --  positive definite matrix A.

Line 109 (Lemma 3): The definition of the bracket operator [] is missing.

Pages 6, 7, ...: The operators col, diag, e should be set in roman type,  not italics -- for example, "sym" in (20) it is correct.

Page 9 - lines 195-196: Prove that the function f satisfies Assumption 1 (Line 101).

Page 7: The paper uses strange and unusual symbols instead of letters - line 144, page 6--(19).

Page 8--line 161: "the error (7) is stable" -- incorrect reference.

Typos, grammatical errors, and spelling mistakes:  
--------------------------------------------------

Line 101: "matrixs" - matrices

Line 112: "will be propsed"

Comments on the Quality of English Language

English is basically fine, only minor corrections will be needed.

Round 2

Reviewer 2 Report

Comments and Suggestions for Authors

My concerns were properly addressed/clarified in the revised version of the paper. I have no further major comments or suggestions.

Minor comments:

-----------------------

Page 4: The inequality (8) need a correction. Additionally, I could not find Lemma 3 in the reference [25], please check it, and possibly clarify.

Page 4, Lemma 3: What is the significance of the number "13" in Lemma 3, and elsewhere in the paper? Also, it is not clear what relation the matrices A and B have to the inequality (8). The sign of the "cross" in (8) is not defined. Thus Lemma 3 is completely unintelligible.

Page 7: The equality (28) should be reformatted; and "colorblue".

Page 9 - line 163: There is still a bad reference, (7) instead of (6).

Comments on the Quality of English Language

The English is fine, only minor corrections are needed.
